# What if you were not there? Learning causally-aware representations of multi-agent interactions

## Abstract

Modeling spatial-temporal interactions between neighboring agents is at the heart of multi-agent problems such as motion forecasting and crowd navigation. Despite notable progress, it remains unclear to which extent modern representations can capture the causal relationships behind agent interactions. In this work, we take an in-depth look at the causal awareness of the learned representations, from computational formalism to controlled simulations to real-world practice. First, we cast doubt on the notion of non-causal robustness studied in the recent CausalAgents benchmark (Roelofs et al., 2022). We show that recent representations are already partially resilient to perturbations of non-causal agents, and yet modeling indirect causal effects involving mediator agents remains challenging. Further, we introduce a simple but effective regularization approach leveraging causal annotations of varying granularity. Through controlled experiments, we find that incorporating finer-grained causal annotations not only leads to higher degrees of causal awareness but also yields stronger out-of-distribution robustness. Finally, we extend our method to a sim-to-real causal transfer framework by means of cross-domain multi-task learning, which boosts generalization in practical settings even without real-world annotations. We hope our work provides more clarity to the challenges and opportunities of learning causally-aware representations in the multi-agent context while making a first step towards a practical solution. Our code is available at `https://github.com/socialcausality`.

## 1 Introduction

Modeling multi-agent interactions with deep neural networks has made great strides in the past few years (Alahi et al., 2016; Vemula et al., 2018; Gupta et al., 2018; Kosaraju et al., 2019; Salzmann et al., 2020; Mangalam et al., 2021; Gu et al., 2021; Xu et al., 2022; Chen et al., 2023). Yet, existing representations still face tremendous challenges in handling changing environments: they often suffer from substantial accuracy drops under mild environmental changes (Liu et al., 2021; Bagi et al., 2023) and require a large number of examples for adaptation to new contexts (Moon & Seo, 2022; Kothari et al., 2022). These challenges are arguably rooted in the nature of the learning approach that seeks statistical correlations in the training data, regardless of their stability and reusability across distributions (Castri et al., 2022; Bagi et al., 2023). One promising solution is to build more *causally-aware representations* – latent representations that are capable of capturing the invariant causal dependencies behind agent interactions.

However, discovering causal knowledge from observational data is often exceptionally difficult (Schölkopf et al., 2021; Schölkopf & von Kügelgen, 2022). Most prior works resort to additional information, such as structural knowledge (Chen et al., 2021; Makansi et al., 2021) and domain labels (Liu et al., 2022; Hu et al., 2022; Bagi et al., 2023). While these attempts have been shown effective in certain out-of-distribution scenarios, they still fall short of explicitly accounting for causal relationships between interactive agents. More recently, CausalAgents (Roelofs et al., 2022) made an effort to collect annotations of agent relations (causal or non-causal) in the Waymo dataset (Ettinger et al., 2021), providing a new benchmark focused on the robustness issue under non-causal agent perturbations. Nevertheless, the reason behind the robustness issue remains unclear, as does the potential use of the collected annotations for representation learning.

The goal of this work is to provide an in-depth analysis of the challenges and opportunities of learning causally-aware representations in the multi-agent context. To this end, we first take a critical

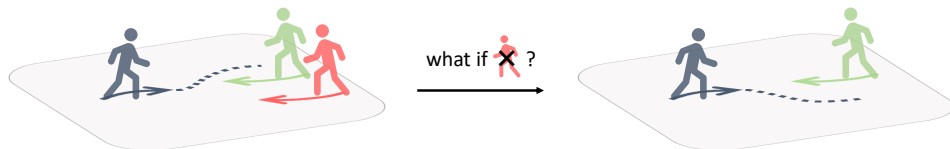

**Figure 1:** Illustration of multi-agent interactions. The behavior of the ego agent is causally influenced by some neighbors. We study the challenges and potential for modeling the causal relations between interactive agents.

look at the recent CausalAgents (Roelofs et al., 2022) benchmark. We find that its labeling procedure and evaluation protocol, unfortunately, present subtle yet critical caveats, thereby resulting in a highly biased measure of robustness. To mitigate these issues, we construct a diagnostic dataset through counterfactual simulations and revisit a collection of recent multi-agent forecasting models. Interestingly, we find that most recent models are already partially resilient to perturbations of non-causal agents but still struggle to capture indirect causal effects that involve mediator agents.

To further enhance the causal robustness of the learned representations, we propose a regularization approach that seeks to preserve the causal effect of each individual neighbor in an embedding space. Specifically, we devise two variants that exploit annotations with different levels of granularity: (i) a contrastive-based regularizer using binary annotations of causal/non-causal agents; (ii) a ranking-based regularizer using continuous annotations of causal effects. Through controlled experiments, we show that both regularizers can enhance causal awareness to notable degrees. More crucially, we find that finer-grained annotations are particularly important for generalization out of the training distribution, such as higher agent density or unseen context arrangements.

Finally, we introduce a sim-to-real causal transfer framework, aiming at extending the strengths of causal regularization from simulation to real-world contexts. We achieve this through cross-domain multi-task learning, *i.e.*, jointly train the representation on the causal task in simulation and the forecasting task in the real world. Through experiments on the ETH-UCY dataset (Lerner et al., 2007; Pellegrini et al., 2010) paired with an ORCA simulator (van den Berg et al., 2008), we find that the causal transfer framework enables stronger generalization in challenging settings such as low-data regimes, even in the absence of real-world causal annotations. As one of the first steps towards causal models in the multi-agent, we hope our work brings new light on the challenges and opportunities of learning causally-aware representations in practice.

## 2 RELATED WORK

The social causality studied in this work is lies at the intersection of three areas: multi-agent interactions, robust representations, and causal learning. In this section, we provide a brief overview of the existing literature in each area and then discuss their relevance to our work.

**Multi-Agent Interactions.** The study of multi-agent interactions has a long history. Early efforts were focused on hand-crafted rules, such as social forces (Helbing & Molnar, 1998; Mehran et al., 2009) and reciprocal collision avoidance (van den Berg et al., 2008; Alahi et al., 2014). Despite remarkable results in sparse scenarios (Luber et al., 2010; Zanlungo et al., 2011; Ferrer et al., 2013), these models often lack social awareness in more densely populated and complex environments (Rudenko et al., 2020). As an alternative, recent years have witnessed a paradigm shift toward learning-based approaches, particularly the use of carefully designed neural networks to learn representations of multi-agent interactions (Kothari et al., 2021). Examples include pooling operators (Alahi et al., 2016; Gupta et al., 2018; Deo & Trivedi, 2018), attention mechanisms (Vemula et al., 2018; Sadeghian et al., 2019; Huang et al., 2019), spatio-temporal graphs (Kosaraju et al., 2019; Salzmann et al., 2020; Li et al., 2020), among others (Rhinehart et al., 2019; Chai et al., 2020; Choi et al., 2021). Nevertheless, the robustness of these models remains a grand challenge (Saadatnejad et al., 2022; Roelofs et al., 2022; Cao et al., 2023). Our work presents a solution to enhance robustness by effectively exploiting causal annotations of varying granularity.

**Robust Representations.** The robustness of machine learning models, especially under distribution shifts, has been a long-standing concern for safety-critical applications (Recht et al., 2019).

Existing efforts have explored two main avenues to address this challenge. One line of work seeks to identify features that are invariant across distributions. Unfortunately, this approach often relies on strong assumptions about the underlying shifts (Hendrycks* et al., 2020; Liu et al., 2021) or on access to multiple training domains (Arjovsky et al., 2020; Krueger et al., 2021), which may not be practical in real-world settings. Another approach aims to develop models that can efficiently adapt to new distributions by updating only a small number of weight parameters, such as sub-modules (Kothari et al., 2022), certain layers (Lee et al., 2023), or a small subset of neurons (Chen et al., 2022). More recently, there has been a growing interest in exploring the potential of causal learning to address the robustness challenges (van Steenkiste et al., 2019; Dittadi et al., 2020; Montero et al., 2022; Dittadi et al., 2022). To the best of our knowledge, our work makes the first attempt in the multi-agent context, showcasing the benefits of incorporating causal relationships for stronger out-of-distribution generalization in more crowded spaces.

**Causal Learning.** Empowering machine learning with causal reasoning has gained a growing interest in recent years (Schölkopf, 2019; Schölkopf & von Kügelgen, 2022). One line of work seeks to discover high-level causal variables from low-level observations, *e.g.*, disentangled (Chen et al., 2016; Higgins et al., 2017; Locatello et al., 2019) or structured latent representations (Locatello et al., 2020; Schölkopf et al., 2021). Unfortunately, existing methods remain largely limited to simple and static datasets (Liu et al., 2023). Another thread of work attempts to draw causal insights into dynamic decision-making (Huang et al., 2022). In particular, recent works have proposed a couple of methods to incorporate causal invariance and structure into the design and training of forecasting models in the multi-agent context (Chen et al., 2021; Makansi et al., 2021; Liu et al., 2022; Hu et al., 2022; Castri et al., 2022; Bagi et al., 2023). However, these efforts have mainly focused on using causal implications to enhance robustness rather than explicitly examining causal relations between interactive agents. Closely related to ours, another recent study introduces a motion forecasting benchmark with annotations of causal relationships (Roelofs et al., 2022). Our work takes a critical look at its labeling as well as evaluation protocols and designs practical methods to enhance causal awareness.

## 3 FORMALISM

In this section, we seek to formalize the robustness challenge in multi-agent representation learning through the lens of social causality. We will first revisit the design of the CausalAgents (Roelofs et al., 2022) benchmark, drawing attention to its caveats in labeling and evaluation. Subsequently, we will introduce a diagnostic dataset, aiming to facilitate more rigorous development and evaluation of causally-aware representations through counterfactual simulations.

### 3.1 SOCIAL INTERACTION AND SOCIAL CAUSALITY

**Social Interaction.** Consider a motion forecasting problem where an ego agent is surrounded by a set of neighboring agents in a scene. Let $s_t^i = (x_t^i, y_t^i)$ denote the state of agent $i$ at time $t$ and $s_t = \{s_t^0, s_t^1, \cdots, s_t^K\}$ denote the joint state of all the agents in the scene. Without loss of generality, we index the ego agent as 0 and the rest of neighboring agents as $\mathcal{A} = \{1, 2, \cdots, K\}$. Given a sequence of history observations $\boldsymbol{x} = (s_1, \cdots, s_t)$, the task is to predict the future trajectory of the ego agent $\boldsymbol{y} = (s_{t+1}^0, \cdots, s_T^0)$ until time $T$. Modern forecasting models are largely composed of encoder-decoder neural networks, where the encoder $f(\cdot)$ first extracts a compact representation $\boldsymbol{z}$ of the input with respect to the ego agent and the decoder $g(\cdot)$ subsequently rolls out a sequence of its future trajectory $\hat{\boldsymbol{y}}$:

$$\begin{aligned}
\boldsymbol{z} &= f(\boldsymbol{x}) = f(s_{1:t}), \\
\hat{\boldsymbol{y}} &= \hat{s}_{t+1:T}^0 = g(\boldsymbol{z}).
\end{aligned} \tag{1}$$

**Social Causality.** Despite remarkable progress on accuracy measures (Alahi et al., 2016; Salzmann et al., 2020), recent neural representations of social interactions still suffer from a significant robustness concern. For instance, recent works have shown that trajectories predicted by existing models often output colliding trajectories (Liu et al., 2021), vulnerable to adversarial perturbations (Saadatnejad et al., 2022; Cao et al., 2022) and deteriorate under distribution shifts of spurious feature such as agent density (Liu et al., 2022).

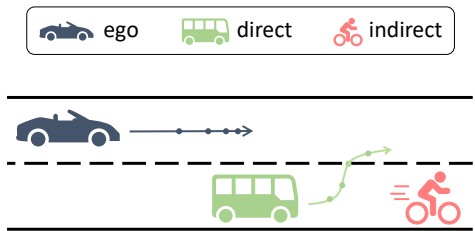

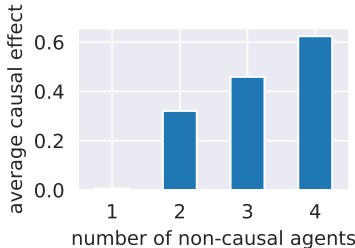

**Figure 2:** Illustration of indirect causal effects. The cyclist *indirectly* influences the decision of the ego agent due to the presence of the bus.

**Figure 3:** Joint effect of all non-causal agents in the scene, simulated in ORCA. The ego agent often changes its behavior when 2+ non-causal neighbors are removed.

One particular notion of robustness that has recently gained much attention is related to the causal relationships between interactive agents (Roelofs et al., 2022). Ideally, neural representations of social interactions should capture the influence of neighboring agents, namely how the future trajectory of the ego agent will vary between a pair of scenes: an original scene $x_\varnothing$ and a perturbed scene $x_R$ where some neighboring agents $\mathcal{R}$ are removed, *i.e.*, only the ego agent and $\mathcal{A} \setminus \mathcal{R}$ remain. We define the causal effect of the removed agents $\mathcal{R}$ as,

$$\mathcal{E}_\mathcal{R} = \|y_\varnothing - y_\mathcal{R}\|_2, \tag{2}$$

where $y_\varnothing \equiv y$ is the future trajectory in the original scene, $y_\mathcal{R}$ is the future trajectory in the perturbed scene, and $\|\cdot\|_2$ is the average point-wise Euclidean distance between two trajectories.

## 3.2 CAVEATS OF SOCIAL CAUSALITY BENCHMARK

While the mathematical definition of the causal effect is straightforward in the multi-agent context, measuring Eq. (2) in practice can be highly difficult. In general, it is impossible to observe a subset of agents replaying their behaviors in the same environment twice in the real world – an issue known as the impossibility of counterfactuals (Peters et al., 2017). To mitigate the data collection difficulty, a recent benchmark CausalAgents (Roelofs et al., 2022) proposes another simplified labeling strategy: instead of collecting paired counterfactual scenes, it queries human labelers to divide neighboring agents into two categories: causal agents that directly influence the driving behavior of the ego agent from camera viewpoints; and non-causal agents that do not. This labeling approach is accompanied by an evaluation protocol through agent removal, assuming that robust forecasting models should be insensitive to scene-level perturbations that remove non-causal agents. More formally, the CausalAgents benchmark evaluates the robustness of a learned representation through the following measure,

$$\Delta = \|\hat{y}_\mathcal{R} - y_\mathcal{R}\|_2 - \|\hat{y}_\varnothing - y_\varnothing\|_2, \tag{3}$$

where $\mathcal{R}$ is the set of all (or some) non-causal agents in the scene.

**Caveats.** In spite of the precious efforts on human annotations, the CausalAgents benchmark (Roelofs et al., 2022) is unfortunately plagued by two subtle but fundamental flaws.

1. *Annotation*: The labeling rule completely overlooks indirect causal relationships, *i.e.*, a neighbor does not directly influence the behavior of the ego agent but does so indirectly by influencing one or a few other neighbors that pass the influence to the ego agent, as illustrated in Fig. 2. On the one hand, indirect causal effects are practically non-trivial for human labelers to identify due to complex relation chains. On the other hand, they are prevalent in densely populated scenes, not only posing a significant modeling challenge but also playing a crucial role in causal learning, which we will demonstrate in §5.

2. *Evaluation*: The evaluation protocol tends to dramatically overestimate the robustness issue. As per Eq. (3), non-causal agents are delineated at the individual level, but robustness assessment occurs at the category level, *e.g.*, removing all non-causal agents rather than a single one. In fact, the joint effect of non-causal agents can escalate quickly with an increasing number of agents, as demonstrated in Fig. 3. Such a discrepancy between annotation and evaluation with respect to non-causal agents can lead to an inflated perception of the robustness issue.

### 3.3 Diagnostic Dataset through Counterfactuals

**Counterfactual Pairs.** To address the above caveats, we create a new diagnostic dataset through counterfactual simulations with ORCA (van den Berg et al., 2008). Specifically, we annotate the ground-truth causal effect by explicitly comparing the trajectories in paired scenes before and after agent removals, as defined in Eq. (2). Details of the collected data are summarized in Appendix B.

**Fine-grained Category.** In addition to real-valued causal effects, we also seek to annotate the category of each agent. Specifically, from simulations, we extract the per-step causal relation between a neighboring agent $i$ and the ego agent. If agent $i$ is visible to the ego at time step $t$, it directly influences the ego, indicated by $\mathbb{1}_t^i = 1$; otherwise $\mathbb{1}_t^i = 0$. We then convert the causal effect over the whole sequence $\mathcal{E}_i$ into three agent categories:

- *Non-causal agent*: little influence on the behavior of the ego agent, *i.e.*, $\mathcal{E} < \epsilon \approx 0$.
- *Direct causal agent*: significant influence on the behavior of the ego agent $\mathcal{E} > \eta \gg 0$; moreover, the influence is direct for at least one time step $\prod_{\tau=1:T}(1 - \mathbb{1}_\tau^i) = 0$.
- *Indirect causal agent*: significant influence on the behavior of the ego agent $\mathcal{E} > \eta \gg 0$; however, the influence is never direct over the entire sequence $\prod_{\tau=1:T}(1 - \mathbb{1}_\tau^i) = 1$.

The collected diagnostic dataset with different levels of causal annotations allows us to rigorously probe the causal robustness of existing representations and to develop more causally-aware representations, which we will describe in the next sections.

## 4 Method

In this section, we will introduce a simple yet effective approach to promote causal awareness of the learned representations. We will first describe a general regularization method, which encapsulates two specific instances exploiting causal annotations of different granularity. We will then introduce a sim-to-real transfer framework that extends the regularization method to practical settings, even in the absence of real-world annotations.

### 4.1 Causally-Aware Regularization

Recall that the causal effect $\mathcal{E}_i$ of an agent $i$ is tied to the *difference* of the potential outcomes between the factual scene and the counterfactual one where the agent $i$ is removed. In this light, a causally-aware representation should also capture such relations, *i.e.*, feature vectors of paired scenes are separated by a certain distance $d_i$ depending on $\mathcal{E}_i$. Motivated by this intuition, we measure the distance between a pair of counterfactual scenes in an embedding space through cosine distance,

$$d_i = 1 - \text{sim}(\boldsymbol{p}_\varnothing, \boldsymbol{p}_i) = 1 - \boldsymbol{p}_\varnothing^\top \boldsymbol{p}_i / \|\boldsymbol{p}_\varnothing\| \|\boldsymbol{p}_i\|, \tag{4}$$

where $\boldsymbol{p}_i = h(\boldsymbol{z}_i) = h(f(\boldsymbol{x}_i))$ is a low-dimensional feature vector projected from the latent representation $\boldsymbol{z}_i$ through a non-linear head $h$. The desired representation is thus expected to preserve the following property,

$$\mathcal{E}_i < \mathcal{E}_j \implies d_i < d_j, \quad \forall i, j \in \mathcal{A}, \tag{5}$$

where $i$ and $j$ are the indices of two agents in the set of neighboring agents $\mathcal{A}$. We next describe two specific tasks that seek to enforce this property Eq. (5), while taking into account the concrete forms of causal annotations with different granularity, as illustrated in Fig. 4.

**Causal Contrastive Learning.** As discussed in §3.3, one simple form of causal annotations is binary labels, indicating whether or not a neighboring agent causally influences the behavior of the ego agent. Intuitively, the representations of paired counterfactual scenes with respect to non-causal agents should be quite different from those with respect to causal ones: the former should stay close to the factual scenes in the embedding space, whereas the latter should be rather far away. We formulate this intuition into a causal contrastive learning objective,

$$\mathcal{L}_{\text{contrast}} = -\log \frac{\exp(d^+/\tau)}{\exp(d^+/\tau) + \sum_k \exp(d_k/\tau)\mathbb{1}_{\mathcal{E}_k > \eta}}, \tag{6}$$

where the positive (distant) example is sampled from counterfactual pairs with respect to causal agents, the negative (nearby) examples are sampled from counterfactual pairs with respect to non-causal agents, $\tau$ is a temperature hyperparameter controlling the difficulty of the contrastive task.

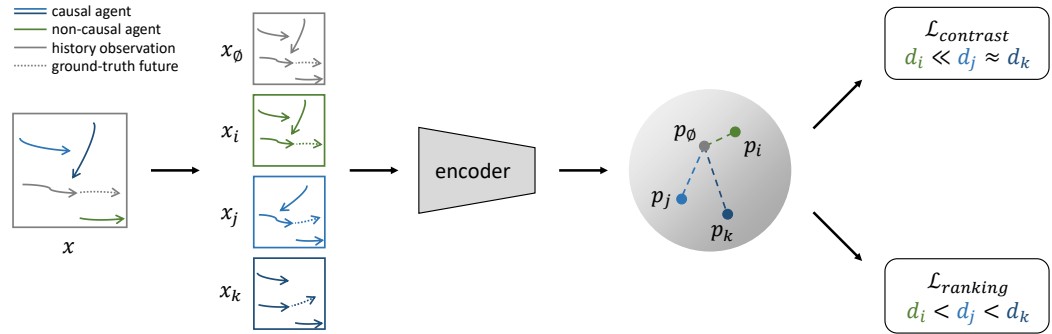

**Figure 4:** Overview of our method. We seek to build an encoder that captures the causal effect of each agent by regularizing the distance of paired embeddings between the factual and counterfactual scenes. We formulate this objective into a contrastive task given binary annotations or a ranking task given real-value annotations.

**Causal Ranking Learning.** One downside of causal contrastive learning described above is that it inherently ignores the detailed effect of causal agents. It tends to push the embeddings of counterfactuals equally far apart across all causal agents, regardless of the variation of causal effects, which violates the desired property stated in Eq. (5). To address this limitation, we further consider another variant of causal regularization using real-valued annotations to provide more dedicated supervision on the relative distance in the embedding space. Concretely, we first sort all agents in a scene based on their causal effect and then sample two agents with different causal effects for comparison. This allows us to formulate a ranking problem in a pairwise manner through a margin ranking loss,

$$\mathcal{L}_{\text{ranking}} = \max(0, d_i - d_j + m), \tag{7}$$

where $d_i$ and $d_j$ are the embedding distances with respect to two agents of different causal effects $\mathcal{E}_i < \mathcal{E}_j$, and $m$ is a small margin hyperparameter controlling the difficulty of the ranking task.

## 4.2 SIM-TO-REAL CAUSAL TRANSFER

The causal regularization method described above relies upon the premise that annotations of causal effects are readily available. However, as elaborated in §3.3, procuring such causal annotations in real-world scenarios can be highly difficult. To bridge this gap, we extend our causal regularization approach to a sim-to-real transfer learning framework. Our key idea is that, despite discrepancies between simulation environments and the real world, some underlying causal mechanisms like collision avoidance and group coordination are likely stable across domains. As such, in the absence of real-world causal annotations, we can instead leverage the causal annotations derived from the simulation counterparts to jointly train the model on the prediction task $\mathcal{L}_{task}^{real}$ on the real-world data and the causal distance task on the simulation data $\mathcal{L}_{causal}^{syn}$,

$$\mathcal{L} = \mathcal{L}_{\text{task}}^{\text{real}} + \alpha \mathcal{L}_{\text{causal}}^{\text{syn}}, \tag{8}$$

where $\alpha$ is a hyperparameter controlling the emphasis on the causal task. Despite its simplicity, we will show in §5.3 that the sim-to-real causal transfer framework effectively translates the causal knowledge absorbed from simulations to real-world benchmarks, *e.g.*, the ETH-UCY dataset (Lerner et al., 2007; Pellegrini et al., 2010), even without any causal annotations from the latter.

## 5 EXPERIMENTS

In this section, we will present a set of experiments to answer the following four questions:

1. How well do recent representations capture the causal relations between interactive agents?
2. Is our proposed method effective for addressing the limitations of recent representations?
3. Do finer-grained annotations provide any benefits for learning causally-aware representations?
4. Finally, does greater causal awareness offer any practical advantages in challenging scenarios?

Throughout our experiments, the multi-agent forecasting task is defined as predicting the future trajectory of the ego agent for 12 time steps, given the history observations of all agents in a scene in the past 8 time steps. We evaluate forecasting models on three metrics:

**Table 1:** Performance of modern representations on the created diagnostic dataset. The prediction errors and causal errors are generally substantial across all evaluated models. However, the errors associated with ACE-NC are rather marginal, suggesting that recent models are already partially robust to non-causal perturbations.

| | ADE ↓ | FDE ↓ | ACE-NC ↓ | ACE-DC ↓ | ACE-IC ↓ |
|---|---|---|---|---|---|
| D-LSTM (Kothari et al., 2021) | 0.329 | 0.677 | 0.027 | 0.532 | 0.614 |
| S-LSTM (Alahi et al., 2016) | 0.314 | 0.627 | 0.031 | 0.463 | 0.523 |
| Trajectron++ Salzmann et al. (2020) | 0.312 | 0.630 | 0.024 | 0.479 | 0.568 |
| STGCNN (Mohamed et al., 2020) | 0.307 | 0.564 | 0.049 | 0.330 | 0.354 |
| AutoBots (Girgis et al., 2021) | 0.255 | 0.497 | 0.045 | 0.595 | 0.616 |

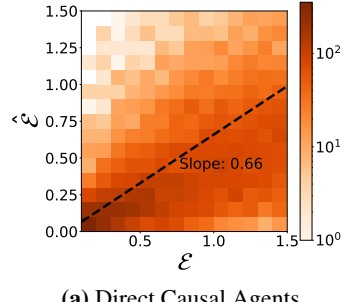

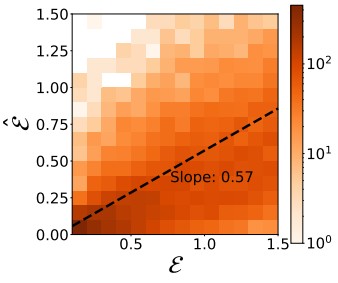

**(a)** Direct Causal Agents        **(b)** Indirect Causal Agents

**Figure 5:** Comparison between estimated causal effects $\hat{\mathcal{E}}$ and the corresponding ground truth $\mathcal{E}$. We uniformly sample the ground-truth causal effect, collect the estimate from AutoBots (Girgis et al., 2021), and linearly regress a slope between the estimated value and the ground truth. The regression slope for a perfect model is 1.

- *Average Displacement Error* (ADE): the average Euclidean distance between the predicted output and the ground truth trajectories – a metric widely used in prior literature to measure the prediction accuracy of a forecasting model.
- *Final Displacement Error* (FDE): the Euclidean distance between the predicted output and the ground truth at the last time step – another common metric measuring prediction accuracy.
- *Average Causal Error* (ACE): the average difference between the estimated causal effect and the ground truth causal effect – a metric inspired by Roelofs et al. (2022) to measure the causal awareness of a learned representation,

$$\text{ACE} \coloneqq \frac{1}{K} \sum_{i=1}^{K} |\hat{\mathcal{E}}_i - \mathcal{E}_i| = \frac{1}{K} \sum_{i=1}^{K} |\|\hat{\boldsymbol{y}}_\varnothing - \hat{\boldsymbol{y}}_i\|_2 - \mathcal{E}_i|. \tag{9}$$

Alongside the aggregated ACE, we also measure causal awareness on each category separately, *i.e.*, ACE-NC, ACE-DC, ACE-IC for non-causal, direct causal, and indirect causal agents, respectively.

## 5.1 ROBUSTNESS OF RECENT REPRESENTATIONS

**Setup.** We start our experiments with the evaluation of a collection of recent models on the diagnostic dataset described in §3.3, including S-LSTM (Alahi et al., 2016), D-LSTM (Kothari et al., 2021), Trajectron++ (Salzmann et al., 2020), STGCNN (Mohamed et al., 2020), and AutoBots (Girgis et al., 2021). Thanks to the fine-grained annotations from counterfactual simulations, we explicitly examine the errors of causal effect estimation for each agent category. We summarize the results in Tab. 1 and visualize the detailed estimation in Fig. 5.

**Takeaway 1: recent representations are already partially robust to non-causal agent removal.** As shown in Tab. 1, the values of ACE-NC are generally quite minimal compared to the results on other metrics. Across all evaluated models, the errors made upon non-causal agents are 10x smaller than that for causal agents (ACE-DC/IC). Corroborate with our analysis in §3.3, this result provides a counterargument to the robustness issue benchmarked in Roelofs et al. (2022), suggesting the importance of studying the robustness under causal perturbations as opposed to non-causal ones.

**Takeaway 2: recent representations underestimate causal effects, particularly indirect ones.** As shown in Fig. 5, the estimate of causal effect often deviates from the ground truth value. In

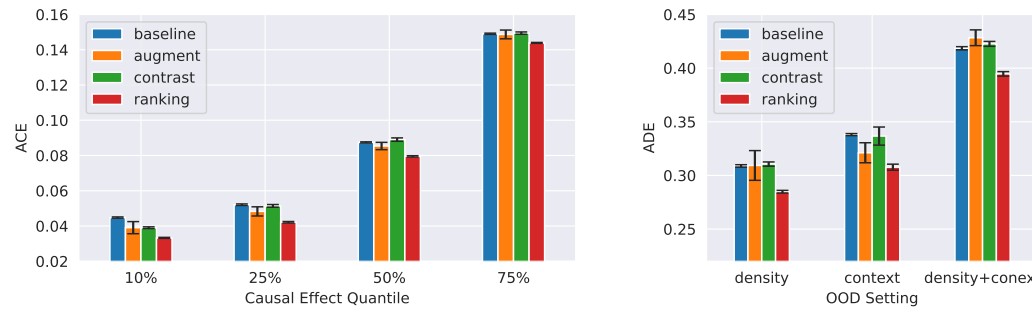

**(a)** In-Distribution Evaluation        **(b)** Out-of-Distribution Evaluation

**Figure 6:** Quantitaive results of our causal regularization method in comparison to the Autobots baseline (Girgis et al., 2021) and the non-causal data augmentation (Roelofs et al., 2022) Models trained by our ranking-based method yield enhanced accuracy in estimating causal effects and more reliable predictions on out-of-distribution test sets. Results are averaged over five random seeds.

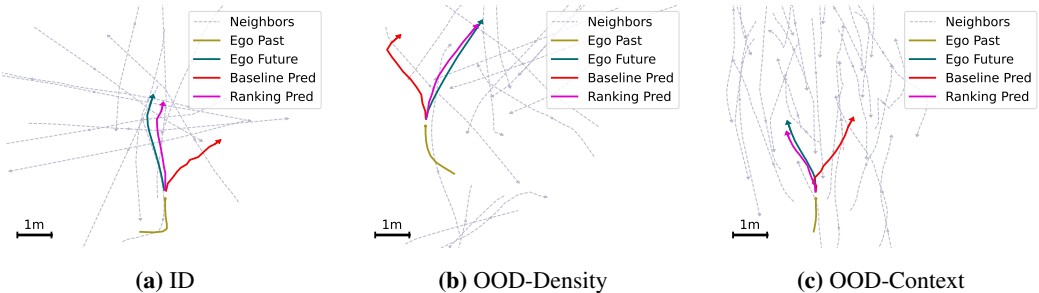

**(a)** ID        **(b)** OOD-Density        **(c)** OOD-Context

**Figure 7:** Qualitative results of our method on in-distribution (ID) and out-of-distribution (OOD) test sets. Models regularized by our ranking-based method demonstrate more robust understanding of agent interactions in scenarios with higher agent density and unseen scene context.

particular, the learned representation tends to severely underestimate the influence of indirect causal agents. This result underscores the limitation of existing interaction representations in reasoning about a chain of causal relations transmitting across multiple agents.

## 5.2 EFFECTIVENESS OF CAUSAL REGULARIZATION

We next evaluate the efficacy of our causal regularization method on Autobots, the strongest baselines in Tab. 1. We take the data augmentation strategy proposed in (Roelofs et al., 2022) as a baseline, and compare different methods in two aspects: in-distribution causal awareness and out-of-distribution generalization.

### 5.2.1 IN-DISTRIBUTION CAUSAL AWARENESS

**Setup.** We first examine the efficacy of our method by measuring the ACE on the in-distribution test set. Since the error of the baseline model varies substantially across different ranges of causal effects (§5.1), we split the test set into four quantile segments. The result is summarized in Fig. 6.

**Takeaway: our method boosts causal awareness thanks to fine-grained annotations.** As shown in Fig. 6a, both the contrastive and ranking variants of our causal regularization approach result in lower causal errors than the vanilla baseline. In particular, the causal ranking method demonstrates substantial advantages over the other counterparts across all quantities of causal effects, confirming the promise of incorporating fine-grained annotations in learning causally-aware representations.

### 5.2.2 OUT-OF-DISTRIBUTION GENERALIZATION

**Setup.** We further examine the efficacy of our method by measuring the prediction accuracy on out-of-distribution test sets. We consider three common types of distribution shifts in the multi-agent setting: higher density, unseen context, and both combined. We summarize the results of different methods on the OOD test sets in Fig. 6b and visualize the difference of prediction output in Fig. 7.

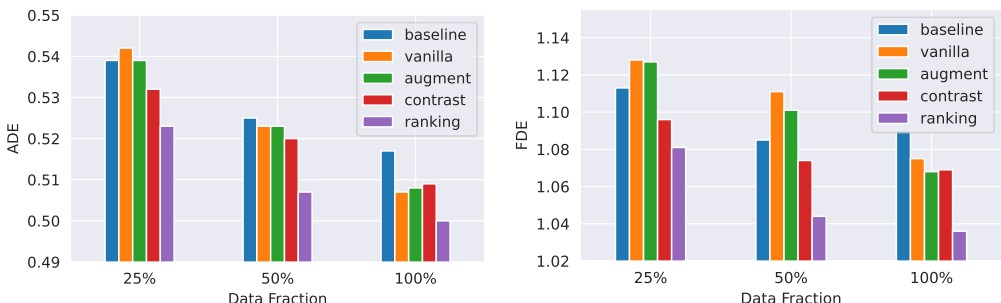

**Figure 8:** Results of causal transfer from ORCA simulations to the ETH-UCY dataset. We consider three settings where the model has access to the same simulation data but varying amounts of real-world data. Our ranking-based causal transfer results in lower prediction errors and higher learning efficiency than the other counterparts, *i.e.*, AutoBots baseline (Girgis et al., 2021), the vanilla sim-to-real transfer (Liang et al., 2020) and the non-causal data augmentation (Roelofs et al., 2022). Details are summarized in Appendices A and B
.

**Takeaway: causally-aware representations lead to stronger out-of-distribution generalization.** As shown in Fig. 6b, the prediction error from our causal ranking method is generally lower than that of the other methods on the OOD sets. In fact, the overall patterns between Fig. 6a and Fig. 6b are highly similar, indicating a substantial correlation between causal awareness and out-of-distribution robustness. This practical benefit is also visually evident in the qualitative visualization in Fig. 7.

## 5.3 EFFECTIVENESS OF CAUSAL TRANSFER

**Setup.** Finally, we evaluate the proposed causal transfer approach on the ETH-UCY dataset (Lerner et al., 2007; Pellegrini et al., 2010) paired with our diagnostic dataset. To assess the potential of the transfer method in challenging settings such as low-data regimes, we train Autobots (Girgis et al., 2021) on different fractions of the real-world data. The results are summarized in Fig. 8.

**Takeaway: causal transfer improves performance in the real world despite domain gaps.** As shown in Fig. 8, our causal transfer approach consistently improves the performance of the learned representation on the real-world benchmark. In particular, it enables the model to learn faster, *e.g.*, even with half data in the real-world, our ranking-based method can still outperform other counterparts. In contrast, the vanilla sim-to-real technique (Liang et al., 2020) fails to yield effective transfer, especially in the low-data regimes. We conjecture that this is due to the disparity between simulation and the real world in terms of non-causal interaction styles, such as speed and curvature.

## 6 CONCLUSIONS

**Summary.** In this paper, we presented a thorough analysis and an effective approach for learning causally-aware representations of multi-agent interactions. We cast doubt on the notion of non-causal robustness in a recent benchmark (Roelofs et al., 2022) and showed that the main weaknesses of recent representations are not overestimations of the effect of non-causal agents but rather underestimation of the effects of indirect causal agents. To boost causal awareness of the learned representations, we introduced a regularization approach that encapsulates a contrastive and a ranking variant leveraging annotations of different granularity. We showed that our approach enables recent models to learn faster, generalize better, as well as transfer stronger to practical problems, even without real-world annotations.

**Limitations.** As one of the first steps towards causally-aware representations in the multi-agent context, our work is subject to two major limitations. On the technical front, while our proposed regularization method consistently boosts causal awareness in various settings, supervision alone is likely insufficient to fully solve causal reasoning in complex scenes, as evidenced by the causal errors in Fig. 6a. Incorporating structural inductive biases might be a promising direction to address high-order reasoning challenges in the presence of indirect causal effects. On the empirical side, while we have demonstrated the potential of our approach in real-world settings in §5.3, most of our other experiments were conducted in controlled simulations for proof-of-concept. Scaling our findings to other practical benchmarks (*e.g.*, WOMD (Ettinger et al., 2021) and INTERACTION (Zhan et al., 2019)) and contexts (*e.g.*, navigation (Chen et al., 2019) and manipulation (Zhou et al., 2023)) can be another fruitful avenue for future research.

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

**Table 2:** Quantitative results of sim-to-real transfer from ORCA simulations to 25% of the ETH-UCY dataset.

| 25% | ETH | HOTEL | UNIV | ZARA1 | ZARA2 | AVG |
|---|---|---|---|---|---|---|
| AutoBots (Girgis et al., 2021) | 0.942/1.886 | 0.333/0.662 | 0.563/1.156 | 0.446/0.958 | 0.409/0.904 | 0.539/1.113 |
| + Vanilla (Liang et al., 2020) | 0.956/1.893 | 0.352/0.698 | **0.537/1.143** | 0.439/0.969 | 0.426/0.935 | 0.542/1.128 |
| + Augment (Roelofs et al., 2022) | 0.942/1.867 | 0.359/0.722 | 0.544/1.152 | 0.441/0.971 | 0.409/0.922 | 0.539/1.127 |
| + Contrast (ours) | 0.938/1.885 | **0.320/0.616** | 0.565/1.187 | 0.450/0.971 | 0.386/0.821 | 0.532/1.096 |
| + Ranking (ours) | **0.920/1.836** | 0.326/0.654 | 0.555/1.144 | **0.438/0.945** | **0.376/0.824** | **0.523/1.081** |

**Table 3:** Quantitative results of sim-to-real transfer from ORCA simulations to 50% of the ETH-UCY dataset.

| 50% | ETH | HOTEL | UNIV | ZARA1 | ZARA2 | AVG |
|---|---|---|---|---|---|---|
| AutoBots (Girgis et al., 2021) | 0.940/1.883 | 0.326/0.612 | 0.566/1.197 | 0.434/0.945 | 0.358/0.787 | 0.525/1.085 |
| + Vanilla (Liang et al., 2020) | 0.923/1.913 | 0.342/0.661 | **0.535/1.141** | 0.430/0.954 | 0.383/0.886 | 0.523/1.111 |
| + Augment (Roelofs et al., 2022) | 0.937/1.885 | 0.340/0.660 | 0.535/1.146 | 0.424/0.938 | 0.377/0.878 | 0.523/1.101 |
| + Contrast (ours) | 0.935/1.870 | 0.344/0.667 | 0.554/1.148 | **0.422/0.913** | 0.346/0.772 | 0.520/1.074 |
| + Ranking (ours) | **0.915/1.826** | **0.303/0.567** | 0.549/1.159 | 0.427/0.919 | **0.340/0.748** | **0.507/1.044** |

**Table 4:** Quantitative results of sim-to-real transfer from ORCA simulations to 100% of the ETH-UCY dataset.

| 100% | ETH | HOTEL | UNIV | ZARA1 | ZARA2 | AVG |
|---|---|---|---|---|---|---|
| AutoBots (Girgis et al., 2021) | 0.938/1.916 | 0.334/0.678 | 0.550/1.152 | 0.420/0.916 | 0.343/0.787 | 0.517/1.090 |
| + Vanilla (Liang et al., 2020) | 0.923/1.913 | 0.331/0.638 | 0.527/1.133 | 0.410/0.907 | 0.346/0.783 | 0.507/1.075 |
| + Augment (Roelofs et al., 2022) | 0.938/1.878 | 0.332/0.635 | **0.518/1.141** | **0.403/0.890** | 0.348/0.796 | 0.508/1.068 |
| + Contrast (ours) | 0.930/1.896 | 0.321/0.605 | 0.538/1.154 | 0.412/0.899 | 0.345/0.791 | 0.509/1.069 |
| + Ranking (ours) | **0.916/1.846** | **0.312/0.582** | 0.539/1.145 | 0.408/0.891 | **0.326/0.714** | **0.500/1.036** |

## A  ADDITIONAL RESULTS

In addition to the aggregated results presented in Fig. 8, we summarize an in-depth breakdown of the sim-to-real transfer results in Tabs. 2 to 4. Across all evaluated settings, our ranking-based causal transfer consistently achieves superior prediction accuracy compared to the AutoBots baseline. Notably, it outpaces the standard sim-to-real method (Liang et al., 2020) in four out of the five subsets, with the sole exception in the UNIV subset. We conjecture that this exception might be attributed to the high similarity between the ORCA simulation and UNIV dataset.

To assess the stability of the ranking-based method, we conducted experiments using three different random seeds, as illustrated in Figs. 11 and 12. The results show that the ranking-based method consistently outperforms the vanilla model across all subsets of the ETH-UCY dataset. Furthermore, the ranking-based method typically yields smaller standard deviations, indicating more consistent performance.

Furthermore, as summarized in Tab. 5, AutoBots stands as one of the state-of-the-art models for multi-agent trajectory forecasting, leaving only marginal room for improvement on the ETH-UCY dataset. In spite of this, our proposed causal transfer method still offers notable improvements, resulting in comparable accuracies to EqMotion (Xu et al., 2023), the most recent model that leverages domain-specific knowledge for predictive tasks.

Finally, we summarize the results of different methods in terms of Final Displacement Error (FDE) on the OOD test sets in Fig. 9. Similar to Fig. 6b, our ranking-based method leads to the lowest prediction errors compared to the other counterparts, reaffirming its strength for boosting out-of-distribution robustness.

## B  IMPLEMENTATION DETAILS

**Experiment details.**  Our experiments are largely built upon the public code of prior work, with as few modifications as possible made for the implementations of our proposed regularizers. Concretely, in the robustness analysis reported in Tab. 1, we train each model on our constructed dataset using the default hyperparameters of the corresponding baseline. To understand the performance of our proposed methods in §5.2, we fine-tune the pre-trained checkpoint for 10 epochs, and evaluate the obtained model on the hold-out test set. The main hyperparameters used for training our baseline model AutoBots (Girgis et al., 2021) and the causal regularizers are listed in Tab. 6.

**Table 5:** Comparison between different multi-agent forecasting models on the ETH-UCY dataset. Boosted by our proposed ranking-based causal transfer, the best result of AutoBots across three random seeds reaches comparable performance to the current state-of-the-art.

| Deterministic | ETH | HOTEL | UNIV | ZARA1 | ZARA2 | AVG |
|---|---|---|---|---|---|---|
| S-LSTM (Alahi et al., 2016) | 1.09/2.35 | 0.79/1.76 | 0.67/1.40 | 0.47/1.00 | 0.56/1.17 | 0.72/1.54 |
| D-LSTM (Kothari et al., 2021) | 1.05/2.10 | 0.46/0.93 | 0.57/1.25 | 0.40/0.90 | 0.37/0.89 | 0.57/1.21 |
| Trajectron++ (Salzmann et al., 2020) | 1.02/2.00 | 0.33/0.62 | 0.53/1.19 | 0.44/0.99 | 0.32/0.73 | 0.53/1.11 |
| EqMotion(Xu et al., 2023) | 0.96/1.92 | **0.30/0.58** | **0.50/1.10** | **0.39/0.86** | **0.30/0.68** | **0.49**/1.03 |
| AutoBots (Girgis et al., 2021) | 0.93/1.87 | 0.32/0.65 | 0.54/1.15 | 0.42/0.91 | 0.34/0.77 | 0.51/1.07 |
| AutoBots + Ranking (ours) | **0.90/1.81** | **0.30/0.56** | 0.53/1.12 | 0.41/0.89 | 0.32/0.71 | **0.49/1.02** |

**Simulation details.** Our diagnostic dataset is generated using a customized version of the Reciprocal Velocity Obstacle simulator that employs Optimal Reciprocal Collision Avoidance (ORCA) (van den Berg et al., 2011). To simulate realistic causal relationships between agents, we imposed a visibility constraint where an agent only observes other neighbors within their proximity and its $210°$ field of view. This visibility plays a significant role in determining the influence of one agent on another. Specifically, we define a neighbor $i$ as having a *direct influence* on the ego agent at time step $t$ if it is visible to the ego in that time step, *i.e.*, $\mathbb{1}_t^i = 1$. Additionally, we introduce a visibility window that records agents that were previously visible, facilitating the modeling of a richer spectrum of direct and indirect inter-agent influences. To encourage the presence of non-causal agents in dense spaces, we explicitly directed specific agents to follow others, thereby making them non-causal or indirect causal.

**Dataset details.** Tab. 7 summarize the key statistics of our diagnostic datasets, including both the in-distribution (ID) training set and the out-of-distribution (OOD) test set. Specifically, we consider three distinct types of OOD datasets, each deviating from the ID dataset in specific aspects, such as agent density and/or scene context.

- *ID*: The training dataset is characterized by an average of 12 pedestrians per scene, each interacting with a few others to navigate towards their goals. All the scenes are set in an open area context, allowing unrestricted movements and serving as the base environment in our experiments.
- *OOD Density*: In our first OOD set, we retain the same context setting as the ID dataset but increase agent density. Specifically, we introduce more agents in proximity to the ego agent to intensify agent interactions. Additionally, we add agents behind the ego agent, which results in more non-causal agents. This dataset aims to test the robustness of the model in handling increased agent density.
- *OOD Context*: The second OOD set alters the scene context from an open area to a narrow street, where pedestrians walk from one end to the other. Given that agents walking in the same direction generally do not interact, we double the number of agents in the scene, thus ensuring a similar degree of interaction complexity to the ID dataset.
- *OOD Density+Context*: The last OOD set differs from the training one in both density and context. Here, we introduce static agents into an open area, akin to a plaza where people occasionally engage in conversations or observe their surroundings. This new type of agents, along with the increased density, presents a challenging OOD setting.

Exemplary animations for each data split can be found in our public repository.

**Baseline details.** To the best of our knowledge, few prior work studies causally-aware representation learning in the multi-agent context. To examine the efficacy of our proposed methods, we consider the following three existing methods as comparative baselines.

- *Baseline*: the baseline method trains the model on the data in the target domain only, *i.e.*, the AutoBots baseline trained on the in-distribution dataset in §5.2 or the real-world ETH-UCY dataset in §5.3.
- *Vanilla*: the vanilla sim-to-real method combines simulated and real-world data in the training process. It adheres to a standard prediction task, with an equal mix of data from each domain in every training batch (Liang et al., 2020).

- *Augment*: the causal augmentation method is built upon the *Baseline* for the experiment in §5.2 or the *Vanilla* for the experiment in §5.3. It augments training data by randomly dropping non-causal agents based on the provided annotations (Roelofs et al., 2022).

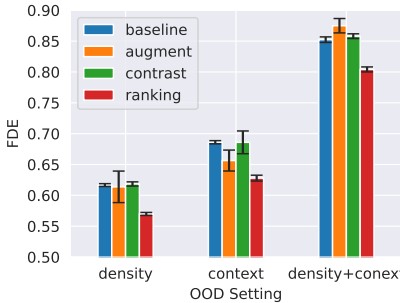

**Figure 9:** Additional quantitative results of our method on the out-of-distribution test sets, as a supplement to Fig. 6b. Models trained by our method yield lower FDE. Results are averaged over five random seeds.

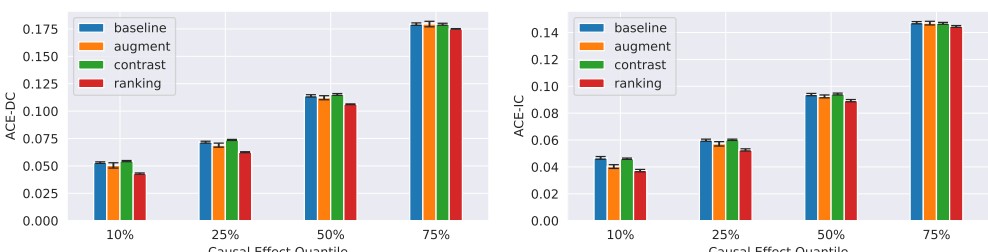

**Figure 10:** Additional quantitative results of our method on causal effect estimates, as a supplement to Fig. 6a. Models trained by our method yield lower ACE-DC and ACE-ID, especially noticeable in scenarios with low-quantile ground-truth causal effects. Results are averaged over five random seeds.
.

## C ADDITIONAL DISCUSSIONS

**Counterfactual simulation.** Our diagnostic dataset, enabled by counterfactual simulations, offers clean annotations of causal relationships, serving as a crucial step in understanding causally-aware representation of multi-agent interactions. However, the realism of these simulated causal effects is still subject to some inherent limitations. For example, we have enforced a stringent constraint on the field of view for each agent, considering that the ego agent is usually unaffected by trailing neighbors. Such constraints could compromise the optimality of the ORCA algorithm, potentially resulting in unnatural trajectories. We believe that integrating more advanced simulators, *e.g.*, CausalCity (McDuff et al., 2022), can address these challenges and we anticipate promising outcomes along this line for future research.

**Multi-agent causal effects.** Our annotation and evaluation have been focused on the causal effect at an individual agent level, namely we remove only one agent at a time. Through this lens, we observe that while recent representations are already partially robust to non-causal agent removal, they tend to underestimate the effects of causal agents. However, it is worth noting that this is still a rather simplified and restricted setting compared to the group-level causal effects, where the collective behavior of multiple agents may have a more complex influence on the ego agent. Understanding and addressing this challenge can be another exciting avenue for future research.

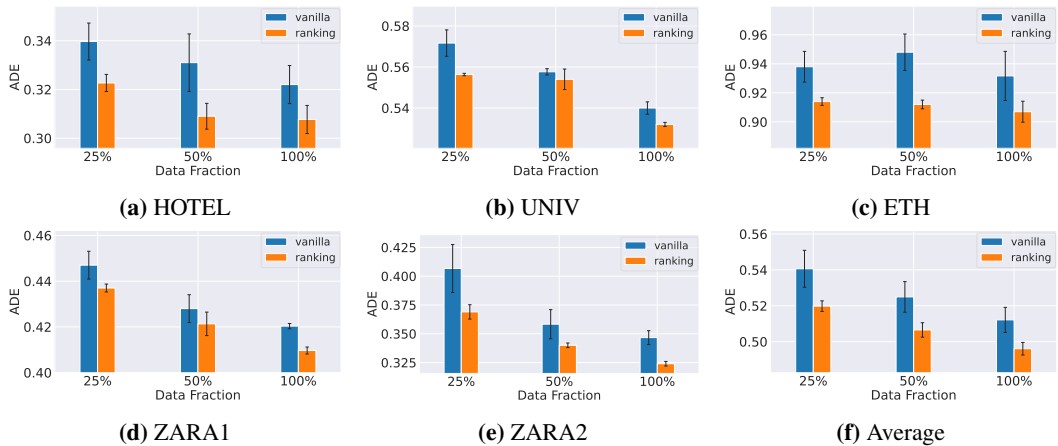

**Figure 11:** Additional results of our ranking-based causal transfer on the ETH-UCY dataset, as a supplement to Fig. 8. The results of ADE are averaged on each subset over three random seeds.

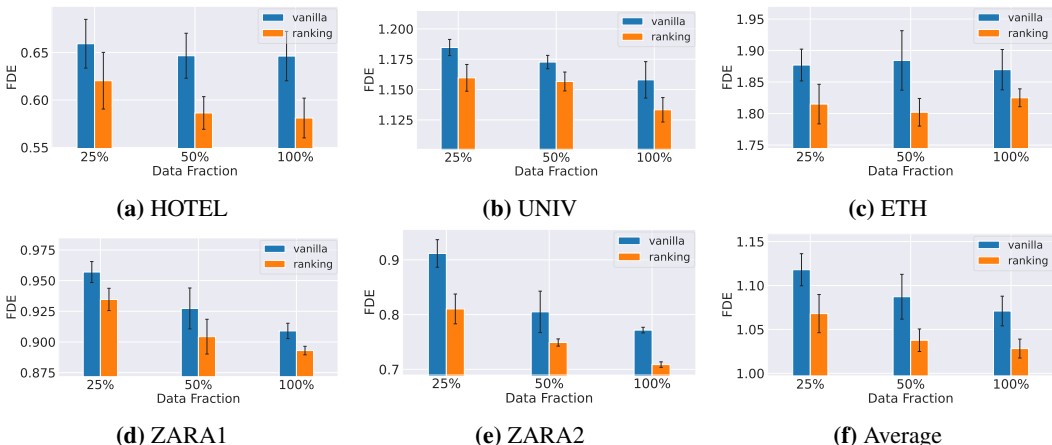

**Figure 12:** Additional results of our ranking-based causal transfer on the ETH-UCY dataset, as a supplement to Fig. 8. The results of FDE are averaged on each subset over three random seeds.

**Table 6:** Key hyper-parameters in our experiments.

| name | value |
|---|---|
| batch size | 16 |
| pre-training learning rate | $7.5 \times 10^{-4}$ |
| fine-tuning learning rate | $2.34375 \times 10^{-5}$ |
| contrastive weight $\alpha$ | 1000 |
| ranking weight $\alpha$ | 1000 |
| ranking margin $m$ | 0.001 |
| non-causal threshold $\epsilon$ | 0.02 |
| causal threshold $\eta$ | 0.1 |

**Table 7:** Key statistics of our diagnostic datasets.

| Dataset | Context | Number of scenes | | Number of agents per scene | | | |
|---|---|---|---|---|---|---|---|
| | | train | test | non-causal | direct causal | indirect causal | total |
| ID | open area | 20k | 2k | 1.31 | 8.35 | 0.48 | 12.03 |
| OOD Density | open area | - | 2k | 9.47 | 12.27 | 2.55 | 28.98 |
| OOD Context | street | - | 2k | 8.23 | 12.22 | 3.09 | 29.01 |
| OOD Density+Context | plaza | - | 2k | 7.49 | 14.64 | 2.78 | 28.93 |

