# OpenReview forum: "What If You Were Not There? Learning Causally-Aware Representations of Multi-Agent Interactions"
_ICLR.cc/2024/Conference — Submitted to ICLR 2024_

### Official Review · Reviewer_FV83 · 2023-10-31

**Soundness:** 4 excellent
**Presentation:** 3 good
**Contribution:** 3 good
**Rating:** 5
**Confidence:** 4

**Summary:**

This paper investigates the extent to which modern representations can capture the causal relationships in agent interactions. It challenges the concept of causal robustness proposed by a recent work. With a new diagnose dataset, this paper shows that recent representations are partially resilient to non-causal agent perturbations but struggle with modeling indirect causal effects involving mediator agents. The study introduces two simple regularization approaches using causal annotations of varying granularity, which enhances causal awareness and out-of-distribution robustness. Additionally, they tested the effectiveness of the proposed method in a sim-to-real causal transfer scenario. This work aims to shed light on the challenges and opportunities in learning causally-aware representations in multi-agent scenarios and takes a step toward practical solutions.

**Strengths:**

(1) The definitions for social causality and the following computational methods are clearly defined and well motived. The overall logic flow is smooth and is a pleasure to read.
(2) The experiments evaluations are thorough.
(3) The writing is clear and helps the understanding of the motivation, desiderata, solution.

**Weaknesses:**

(1) The proposed method is highly confined to the simulator. As the author already noted, counterfactual data is impractical to collect in real world scenario and they instead took a sim-to-real transfer learning approach. But the assumption that the underlying causal mechanisms stay the same is simply too fragile to be true. For example, in simulation, the author define the visibility range as 210 degree in front of the driver ignoring the fact that drivers also check rearview mirrors for road information. And the simulator itself focuses heavily on collision avoidance. However, human drivers are not perfectly rational and have various driving preferences. Instead of “causally-aware” representations, it’s more accurate to say the proposed method learns “collision-aware representations”. Rather than assuming “the causal mechanisms” satisfying some vague constraints, I would recommend the authors model the problem more rigorously with causal theories [1,2].

(2) Even if we only consider the collision-free driving style, the proposed method’s effectiveness is not convincing enough. Firstly, if you have done multiple runs with different random seeds for the same experiment, I am expecting to see confidence intervals or error bars in the results. Secondly, the author introduced three metrics for evaluation, Average Displacement Error (ADE), FinalDisplacementError(FDE), and Average Causal Error (ACE). However, figure 6 is missing FDE and figure 8 is missing ACE. Thirdly, for the data reported, the contrastive method seems to be on par with the “augment”. When in sim-to-real, it’s sometimes even worse than “augment” and the improvement is rather marginal. We do see notable improvements of the “rank” method over baselines but I am not sure if it’s a fair comparison because the “rank” method actually provides way more labeling information than binary labels (causal/non-causal). Does those baselines also have access to such labels? At last, when using 100% percent data in the sim-to-real test, the proposed method (contrast) barely beats those baselines leaving doubts on whether the performance gaps in 25% and 50% scenarios are due to learning efficiency difference instead of representation quality.

[1] Bareinboim, Elias, and Judea Pearl. "Causal inference and the data-fusion problem." Proceedings of the National Academy of Sciences 113.27 (2016): 7345-7352.
[2] Hernán, Miguel A., and Tyler J. VanderWeele. "Compound treatments and transportability of causal inference." Epidemiology (Cambridge, Mass.) 22.3 (2011): 368.

**Questions:**

(1) Could you extend the experiments to multiple random seeds and report the variances in the figures?
(2) Could you report all three metrics for figure 6 and 8?

---

> ### Author Response · Authors · 2023-11-18
>
> Dear Reviewer FV83,
>
> Thank you for your insightful feedback. Please find our response to your comments below.
>
> > The assumption that the underlying causal mechanisms stay the same between the simulation and real-world is simply too fragile to be true. For example, ignoring the fact that drivers also check rearview mirrors.
>
> * We acknowledge your concern and clarify that we do not assume all causal mechanisms stay the same. Instead, we assume they differ between the simulation and real-world in a sparse way (cf. sparse mechanism shift [1,2]). For the mechanisms shared across the two domains, we can first annotate them in the simulation and further transfer them to the real-world.
> * This assumption aligns well with our considered setting, where the simulator is designed and calibrated to mimic pedestrian behaviors (not drivers) in the real-world, including a 210-degree field of view to match human visual perception [3].
> * To the best of our knowledge, existing research in causal representation learning is still confined to synthetic/simulated settings, and struggles to scale to the real-world. Our proposed sim-to-real causal transfer provides a new perspective to bridge this gap and may hold great promise in broader settings beyond the considered application.
>
> > Could you report all three metrics for figure 6 and 8?
> * We’d like to clarify that it is not possible to evaluate ACE for Figure 8, due to the absence of causal annotations in the real-world ETH-UCY dataset.
> * We have added FDE for Figure 6 into the appendix (Figure 9). We didn’t report it in the previous version, because the pattern of FDE is similar to that of ADE and does not provide much additional information.
>
> > We do see notable improvements of the “rank” method over baselines but I am not sure if it’s a fair comparison because the “rank” method actually provides way more labeling information than binary labels (causal/non-causal).
> * We’d like to clarify that the goal of the experiment is not about which method is better given the same annotations. Instead, we aim to examine the relative benefits of fine-grained annotations over coarse-grained labels, as noted at the beginning of the experiment section (Q3).
> * Our result shows that learning from fine-grained annotations through a ranking regularizer yields substantial improvements over the other counterparts. This result provides valuable guidance for causal annotation strategies in simulation and real-world scenarios.
>
> > At last, when using 100% percent data in the sim-to-real test, the proposed method (contrast) barely beats those baselines leaving doubts on whether the performance gaps in 25% and 50% scenarios are due to learning efficiency difference instead of representation quality.
> * We appreciate your observation regarding the performance gaps in different data scenarios. We’d like to clarify that we do not claim superiority of our contrastive variant over the augment baseline; their performance is comparable across most experiments. The main finding we’d like to highlight is the strength of our ranking-based variant, including its high learning efficiency in the low data regime, e.g., reaching similar or even better accuracies with 1/2 less of the real-world data.
>
> > Could you extend the experiments to multiple random seeds and report the variances in the figures?
>
> Thank you for the suggestion.
> * We have updated Figure 6 in our manuscript to report the variance across 5 random seeds.
> * We are still running the experiments on different seeds for Figure 8. Below are the preliminary results on the full (100%) ETH-UCY dataset. We anticipate completing the experiments by next week and will update Figure 8 accordingly.
>
> |             | ETH                       | Hotel                     | Univ                      | Zara1                     | Zara2                     |
> | ----------- | ------------------------- | ------------------------- | ------------------------- | ------------------------- | ------------------------- |
> | Baseline    | 0.932±0.02 / 1.870±0.032  | 0.322±0.008 / 0.646±0.026 | 0.540±0.005 / 1.154±0.006 | 0.420±0.002 / 0.909±0.006 | 0.344±0.004 / 0.766±0.004 |
> | Ranking | 0.907±0.007 / 1.825±0.014 | 0.308±0.006 / 0.581±0.021 | 0.534±0.001 / 1.143±0.001 | 0.410±0.002 / 0.893±0.003 | 0.327±0.001 / 0.718±0.005 |
>
>
> [1] Towards Causal Representation Learning, 2021 \
> [2] From Statistical to Causal Learning, 2022 \
> [3] Seven Myths on Crowding and Peripheral Vision, 2020

---

> ### Comment · Reviewer_FV83 · 2023-11-21
>
> Thank you for your detailed response. Most of my concerns have been clarified. I am looking forward to more experiment results about Fig. 8. However, I would like to pose three follow up questions based on your response.
>
> 1. From the current table, it seems that "ranking" is still not completely beating the baselines, which casts doubts on "the strength of our ranking-based variant". Is this a contradiction to your claim?
>
> 2. Though the references the authors provided are not formatted, I managed to find the papers, which I believe should be the right ones. It seems that the sparse mechanism shift is defined under the Causal Markov Condition, which basically means that there will be no unobserved confounders. I am not sure if it is realistic to model the pedestrian behaviors assuming that there are no spurious correlations among them. And this question could have been better clarified if the author had defined the type of causal model they were using exactly with the formal causal language [1].
>
> 3. If the dataset is all about pedestrian behaviors, Fig 2 seems to be misleading since in that figure the ego agent is a car instead of a pedestrian.

---

> > ### Author Response · Authors · 2023-11-23
> >
> > Dear FV83,
> >
> > Thank you for your detailed feedback on our response.
> >
> > > From the current table, it seems that "ranking" is still not completely beating the baselines
> >
> > * We appreciate your attention to the detailed results in our table. To clarify, our proposed ranking method yields consistent improvement over the baseline. In particular, the Autobots [1] trained by our method have shown comparable performance to the current state-of-the-art, EqMotion [2], on the ETH-UCY dataset, as detailed in Table 5 of our manuscript.
> > * Additionally, we expand our evaluation to include training on various proportions (25%, 50%, 100%) of the ETH-UCY dataset. The results, averaged over three random seeds and summarized in Fig 11&12 of our updated manuscript, corroborate with our findings earlier: the proposed ranking-based causal transfer significantly boosts sample efficiency, enabling the model to reach comparable or even better accuracy with significantly less real-world data.
> >
> > > It seems that the sparse mechanism shift is defined under the Causal Markov Condition, which basically means that there will be no unobserved confounders. I am not sure if it is realistic to model the pedestrian behaviors assuming that there are no spurious correlations among them.
> >
> > * To our knowledge, while the principle of sparse mechanism shift was derived from the distribution of the observables, its implication lies in more general properties of distribution shifts: `real-world distribution as a product of causal mechanisms. A change in such a distribution (e.g., when moving from one domain to a related one) … should usually not affect all mechanisms simultaneously` [3].
> > * We hypothesize that this principle applies to our considered context, even in the presence of unobserved confounders. In fact, we explicitly motivated our approach from the perspective of spurious correlations, as detailed in the social causality paragraph in Section 3.1, and designed OOD settings (e.g., density increase) to test this hypothesis.
> > * Finally, as stated in our manuscript, we consider our work as a first step towards causal representations of multi-agent interactions in practical settings. We acknowledge that our work is still subject to several limitations, e.g., a gap between our proposed sim-to-real causal transfer method and more formal causal modeling of the domain shift. Addressing these limitations, including formalizing and modeling indirect causal effects that existing representations still struggle to capture, can be fascinating directions for future research.
> >
> > > Fig 2 seems to be misleading since in that figure the ego agent is a car instead of a pedestrian
> >
> > * We acknowledge your concern about potential misinterpretation. However, we would like to clarify that our objective here is to explore causally-aware representations in a broad multi-agent context, irrespective of agent types. In this light, we have set the problem context in Fig 1 through an example of pedestrian interactions. In Fig 2, we seek to provide a motivating example of indirect causal relations, where we find differentiating agent types aids in visual clarity. Nevertheless, we are open to suggestions for alternative illustrations and are willing to make necessary adjustments to better convey the message.
> >
> > We hope this clarifies our contributions and welcome any additional feedback you may have.
> >
> > [1] Latent Variable Sequential Set Transformers for Joint Multi-Agent Motion Prediction, ICLR, 2022 \
> > [2] EqMotion: Equivariant Multi-agent Motion Prediction with Invariant Interaction Reasoning, CVPR, 2023 \
> > [3] Toward Causal Representation Learning, Proc. of the IEEE, 2021

---

> ### Comment · Reviewer_FV83 · 2023-11-23
>
> Regarding my point 2 previously, modeling the causal effect rigorously should be the foundation of the problem instead of a “fascinating future work”. And if you check the references you provided, they clearly stated in the context that they described the sparse mechanism shift under the markovian causal model, where there is no confounders.  Citing another vague statement does not help in this case since there are already established causal literatures solving this exactly. You may refer to the line of transportability in causal inference for more details.
>
> Regarding my point 3, as the authors stated in their previous response, the simulator they used “models the behavior of pedestrians” not “drivers”. If your goal is to “explore casually aware representations in a broad multi agent  setting”, you should include more general simulators instead of a pedestrian simulator and should not assuming that there are no confounders between the agents behavior. Again, this confusion is also partially due to the problem that the authors failed to model the problem using precise causal language properly.
>
> Thus, I’ll keep the score.

---

> ### Author Response · Authors · 2023-11-23
>
> Dear Reviewer FV83,
>
> We would like to reiterate the three main contributions of our submission.
> 1. Reveal the annotation and evaluation issues in the recent large-scale CausalAgent benchmark
> 2. Introduce a causal metric learning approach that boosts causal awareness and out-of-distribution robustness in controlled simulations
> 3. Propose a sim-to-real causal transfer technique that yields encouraging results without relying on real-world annotations and thus holds great promise for practical problems
>
> We acknowledge the limitations of our study, as you rightly pointed out, and have explicitly discussed some of them in the `Limitations` and `Additional Discussions` paragraphs in our manuscript.
>
> Given these contributions and discussions, we respectfully disagree with your conclusion that our submission should be rejected, and we believe that it fills critical knowledge gaps in the field.
>
> Finally, we would like to express our sincere gratitude for the time and effort that you have invested in reviewing our work.

---

### Official Review · Reviewer_1Bm7 · 2023-11-01

**Soundness:** 2 fair
**Presentation:** 3 good
**Contribution:** 3 good
**Rating:** 6
**Confidence:** 3

**Summary:**

The paper considers the problem of learning causally-aware representations in the context of multi-agent interactions. The authors first raise concern about the recently proposed social causality benchmark, pointing out the flaws in the annotation process and evaluation protocol. As a result, the authors introduce a diagnostic dataset based on ORCA, with fine-grained causal annotations where each agent is labeled as non-causal, direct causal, or indirect causal, and propose a new evaluation metric, namely average causal error (ACE). The paper then proposes two strategies exploiting fine-grained causal annotations: regularization using contrastive loss or ranking loss. For the real-world scenarios where obtaining counterfactual samples is impossible, they introduce sim-to-real causal transfer strategy where the model is jointly trained on the task in hand and its simulation counterparts. The experiments demonstrate the effectiveness of the proposed regularization strategy using fine-grained causal annotations and causal transfer.

**Strengths:**

- The paper is easy to follow and well-written. The problem of learning causally-aware representations is also significant.
- The paper raises concerns about the CausalAgents benchmark, i.e., the annotation process and evaluation protocol. These points seem reasonable to me and serve as a new perspective at minimum. It could also bring further development in the field (e.g., new benchmarks and evaluation metrics).
- Leveraging fine-grained annotations to further improve the robustness of the representations is convincing. The proposed two instantiations are intuitive and reasonable.
- The motivation and idea of sim-to-real causal transfer are convincing and interesting. It is simple and works well (at least in the ETH-UCY dataset).

**Weaknesses:**

- In the annotation process of the proposed dataset, each $i$-th agent is labeled as non-causal, direct causal, or indirect causal based on the influence the ego agent $\mathcal{E}_i$, i.e., measured by removing a *single agent*. However, the causal effect of the agent (to the ego agent) could be different based on the other agent (as Fig 2 illustrates).
    - For example, for indirect causal agents  $i, j$ (i.e., $\mathcal{E}_i, \mathcal{E}_j \gg 0$), it is possible that removing both of them does not influence the ego agent, i.e., $\mathcal{E}\_{ij} \simeq 0$.
    - For another example, it is also possible that $\mathcal{E}\_{ij} \gg 0$ when $\mathcal{E}_i, \mathcal{E}_j \simeq 0$ (e.g., when there are two bikes in the same lane in Fig 2, removing only one of them does not influence the ego agent but it does when removing them together).
- Similarly, the proposed evaluation metric ACE also considers removing only a single agent. It is not clear **why and how this metric serves as a reliable indicator** of the causal awareness and robustness of a learned representation. The authors argued that Eq. 3 using $\mathcal{E}_\mathcal{R}$ *overestimates* the robustness issue as described in Caveats in page 4 and Fig 3. However, one could also view that the proposed metric **ACE may *underestimate* the robustness issue**. This is related to “Takeaway 1” where the authors claim that recent methods are already partially robust w.r.t. non-causal agent removal. But this only considers a single agent removal and therefore may underestimate the robustness issue. It would be appreciated if the authors could provide justification for their annotation process and evaluation metric which is based on the influence measured by removing *a single agent*.
- The paper only considers a single dataset (ETH-UCY), and the proposed regularization strategy is only applied to a single backbone (AutoBots), which makes it difficult to assess its wide applicability. In other words, it is unclear how the proposed regularization strategy would work on other datasets and with other backbone methods.
- The paper lacks a detailed description of the proposed diagnostic dataset based on ORCA (in both the main body and the appendix). For example, what is the difference between OOD-Context and OOD-Density-Context? (They seem to be both dense.) Maybe a figure or illustration would be much appreciated. Also, as the one who is not directly working in this field, it is hard for me to understand the similarity and difference between ORCA simulation and ETH-UCY dataset, and thus it is difficult to assess how reasonable the proposed causal transfer strategy is.

**Justification for the rating**

Despite several concerns I listed above, my initial assessment weighed more on its positive (potential) values. I look forward to the feedback from the authors and discussions with other reviewers.

**Questions:**

My major concerns and questions are listed above. Some minor comments:

- Fig 2 is not color friendly (it is hard to distinguish cyan and blue).
- The paper sometimes interchangeably uses “causal representation” and “causally-aware representation”. They have different meanings in the literature and I think the latter fits the paper.

---

> ### Author Response · Authors · 2023-11-21
>
> Dear Reviewer 1Bm7,
>
> Thank you for your insightful feedback. Please find our response to your comments below.
>
> > It would be appreciated if the authors could provide justification for their annotation process and evaluation metric which is based on the influence measured by removing a single agent.
>
> * Indeed, the causal effect in the multi-agent context can be studied at both the individual agent level (removing one agent at a time) and the group level (removing a group of multiple agents). Our focus on agent-level causal effects was motivated by two factors:
>   * It aligns with the notion of Causal Agents, i.e., a neighboring agent has a certain causal relationship with the ego agent. This allows us to revisit the root cause of the robustness issues in the prior benchmark.
>   * It permits efficient experiments, i.e., studying causal effects at the agent level presents a linear computational complexity O(N), whereas studying that at the group level could lead to an exponential increase in complexity, up to O(2^N).
> * Having said that, we acknowledge that exploring the latter can be an exciting avenue for future research. We have updated our manuscript to reflect this perspective, framing our finding about `partially robust` in the context of agent-level causal effects and discussing its limitation in Appendix C.
>
> > How the proposed regularization strategy would work on other datasets and with other backbone methods.
>
> * We are currently running experiments on another backbone (D-LSTM) benchmarked in Tab 1. Our preliminary result shows larger performance gains on D-LSTM than on AutoBots. We anticipate completing the experiments soon and will update our response accordingly.
>
> > Lacks a detailed description of the proposed diagnostic dataset. For example, what is the difference between OOD-Context and OOD-Density-Context? Maybe a figure or illustration would be much appreciated.
>
> * The key difference between OOD-Context and OOD-Density-Context is that the former introduces major changes in scene contexts, whereas the latter introduces more significant increases in the number of causal agents as well as modest changes in scene contexts.
> * As per your suggestion, we have added a more detailed description in the `dataset details` paragraph in Appendix B, along with animations in [our public repository](https://github.com/socialcausality/socialcausality#illustrations) for better illustration.
>
> > Fig 2 is not color friendly (it is hard to distinguish cyan and blue).
>
> * Thank you for bringing this to our attention. We have updated the figure, changing blue to red.
>
> > The paper sometimes interchangeably uses “causal representation” and “causally-aware representation”. I think the latter fits the paper.
>
> * Thank you for the suggestion. We have revised the manuscript, using the term “causally-aware representation” more uniformly across the paper.

---

### Official Review · Reviewer_d8CZ · 2023-11-04

**Soundness:** 3 good
**Presentation:** 4 excellent
**Contribution:** 3 good
**Rating:** 6
**Confidence:** 4

**Summary:**

This work focuses on the problem of causal representations in multi-agent forecasting problems.
After assessing the CausalAgents benchmark and its shortcomings, the authors employ the ORCA simulator to generate counterfactual scenes which results in a diagnostic dataset.
This dataset includes annotations of individual agents' causal effects on the ego agent through the euclidean distance of its trajectory in both scenes.
Two methods for promoting causal awareness are introduced, the basis of which is that latent representations of the ego agent should be invariant to the removal or addition of non-causal agents.
Since this approach heavily relies on the causal annotations, its application to real-world datasets is non-trivial.
To bridge this gap, this work proposes a sim-to-real causal transfer approach where data with annotated causal effects is used in conjunction with real human trajectory data to train a motion forecasting model.
Results are presented on the diagnostic dataset to evaluate the causal awareness of current motion prediction models, and how the proposed approaches affect the causal performance of such models.
Further results are presented to show how the proposed approaches can help current prediction models on real-world datasets using the proposed sim-to-real training.

**Strengths:**

- The literature review is complete and addresses the relevant categories for this work. These include multi-agent interactions, robust representations and causal learning.

- The motivation for this work, presented in Section 3.1 and 3.2, is informative and provides the background for the subsequent sections. I particularly liked section 3.2 which outlined the shortcomings of the CausalAgents Benchmark.

- I thought the experiment showing how promoting causal invariance leads to better prediction accuracy with low amounts of real-world data interesting. This is possibly undersold in the paper, but an interesting result.

**Weaknesses:**

- The main issue I have is that the proposed approach seems to provide marginal improvements across all metrics.
Perhaps this is down to the point made by the authors, that all methods are already pretty strong for non-causal agents, but not so much for causal agents.
In Figure 6, the within quantile difference seems quite small.
I think it would be much more interesting if the ACE metrics in Table 1 were replicated with the proposed approaches, showing how these values (particularly ACE-DC/IC) are presumably improved with the promotion of causal invariance. Why is the split done using quantiles here instead of spliting across the ACE-X metrics?

**Questions:**

- Figure 4 is not referenced anywhere.

- in Figure 8, what is the difference between baseline and vanilla?

- In relation to the previous point, I think all Figure captions should include a legend telling the reader the different variants. For examples, Figure 6 has both a baseline and augment. But the text says that data augmentation is a baseline. This feels a little ambiguous. If I understood correctly, augment is the data augmentation baseline, and baseline is actually vanilla Autobots. Is this correct?

- I wonder if it would be interesting to also look at the number(or percentage) of ego-collisions in the sim-to-real experiments. It could give a better picture on the importance of promoting causal representations.

- What exactly is the unseen context variant in the OOD experiments? This and high density should be clearly defined.

---

> ### Author Response · Authors · 2023-11-22
>
> Dear Reviewer d8CZ,
>
> Thank you for your insightful feedback. Please find our response to your comments below.
>
> > The proposed approach seems to provide marginal improvements across all metrics
>
> * We would like to highlight that our method was evaluated on a state-of-the-art baseline, where significant leaps in performance are inherently challenging. Despite this, our ranking-based method yields ~10% performance gain on several metrics, e.g., ACE in the low-quantile causal effect scenarios and ADE in the OOD density and context settings. Most notably, the final version of our method enables the model to reach better accuracy while requiring only half the real-world data.
>
> > It would be much more interesting if the ACE metrics in Table 1 were replicated with the proposed approaches, showing how these values (particularly ACE-DC/IC) are presumably improved. Why is the split done using quantiles here instead of spliting across the ACE-X metrics?
>
> * The quantile split was chosen to better understand the strengths and limits of the proposed method in causal effect estimates. As shown in Fig 6a, the improvement over ACE is greater with smaller ground-truth causal effects. We conjecture this is in large part due to the imperfect decoding process.
>   * When the causal effect of the neighboring agent is small (affects the ground-truth trajectory only locally), encoding the paired scenes into nearby embeddings results in similar prediction outputs and hence small ACE, canceling out the influence of the decoding error.
>   * In contrast, when the causal effect of the neighboring agent is large (affects the ground-truth trajectory at macro scale), even if encoding the paired scenes into properly separated embeddings, the decoding errors of each trajectory may compound and lead to large ACE.
> * Following your suggestion, we have extended our evaluation to include ACE-DC and ACE-IC metrics. The results are summarized in Figure 10 in our updated manuscript. Our proposed method yields comparable improvements on the two metrics. We conjecture this is because our proposed method treats DC and IC agents uniformly.
>
> > Figure 4 is not referenced anywhere.
>
> * Thank you for bringing it to our attention! In our updated manuscript, we have added reference to Fig 4 in the paragraph right after Eq 5.
>
> > In Figure 8, what is the difference between baseline and vanilla?
>
> * The baseline refers to training the model on real-world data only, whereas the vanilla sim-to-real refers to training the model on simulated and real-world data jointly. In our updated manuscript, we elaborate on the details of each method in the `baseline details` paragraph in Appendix B.
>
> > If I understood correctly, augment is the data augmentation baseline, and baseline is actually vanilla Autobots. Is this correct?
>
> * Yes, the baseline refers to the original Autobots throughout our manuscript, whereas the augment refers to the causal data augmentation technique in Roelofs el al. 2022. We have updated figure captions accordingly in our updated manuscript.
>
> > I wonder if it would be interesting to also look at the number(or percentage) of ego-collisions in the sim-to-real experiments.
>
> * Thank you for the suggestion. Our method does not significantly reduce collision rates compared to the baseline. This is because the AutoBots baseline already exhibits exceptionally low collision rates on the ETH-UCY dataset (0.5%), leaving marginal room for improvements.
>
> > What exactly is the unseen context variant in the OOD experiments?
>
> * The training context comprises a simulated open area with randomized agent directions, whereas the OOD context features a simulated street with constrained agent movements. Please refer to more detailed description in the `dataset details` paragraph in Appendix B, along with animations in [our public repository](https://github.com/socialcausality/socialcausality#illustrations) for illustration.

---

### Public Comment · ~Yuke_Li1 · 2023-11-19
**Questions**

Dear Authors,

I recently came across this paper and found it intriguing, particularly regarding the concept of 'Causally-Aware Representations'. However, I have a few questions that I hope the authors can address to clarify my understanding:

1. **Assumptions Underlying Causally-Aware Representations:**
To my understanding, this work seems to operate under the assumption that causal relationships capture multi-agent interactions. However, I did not find explicit justifications for this foundational assumption.

2. **Difference and Connection with Causal Representation:**
While causal representation is mentioned, I'm unclear on how the proposed method aligns with or differs from it. For instance, the framework's identifiability is not addressed. Moreover, it appears that several key references in causal representation learning are omitted, such as [1,2,3,4,5], which might provide additional context.

3. **Data Sufficiency for Experiments:**
The paper seems to utilize the potential outcome framework, typically requiring interventional data to validate assumed causal relationships. Without this data, the fundamental assumption that interactions can be represented causally might be compromised. The absence of discussion on this point is notable. I guess simulating the trajectories could be helpful in this aspect, such as "CausalCity: Complex Simulations with Agency for Causal Discovery and Reasoning".

References:

[1] Kong, Lingjing, et al. "Partial disentanglement for domain adaptation." International Conference on Machine Learning. PMLR, 2022.

[2] Xie, Shaoan, et al. "Multi-domain image generation and translation with identifiability guarantees." The Eleventh International Conference on Learning Representations. 2023.

[3] Yao, Weiran, et al. "Learning temporally causal latent processes from general temporal data." arXiv preprint arXiv:2110.05428 (2021).

[4] Yao, Weiran, et al. "Temporally disentangled representation learning." Advances in Neural Information Processing Systems 35 (2022): 26492-26503.

[5] Feng, Fan, et al. "Factored adaptation for non-stationary reinforcement learning." Advances in Neural Information Processing Systems 35 (2022): 31957-31971.

---

> ### Author Response · Authors · 2023-11-21
>
> Thank you for your insightful comments. Please find our clarifications below.
>
> > Assumptions Underlying Causally-Aware Representations: causal relationships capture multi-agent interactions
>
> * Our work posits that desired representations of multi-agent interactions should capture the causal relationship between behaviors of interactive agents. For instance, consider a scenario where two agents walk in a line, their similarity in pace is not just a statistical correlation but governed by an asymmetric causal relationship, i.e., the agent in front casually influences the agent behind, but not the other way around. This example highlights our belief in the significance of causal relationships for modeling multi-agent interactions. We are open to further discussion and clarification on this point.
>
> > Difference and Connection with Causal Representation: how the proposed method aligns with or differs from it
>
> * Connection: To our knowledge, the essence of causal representations lies in capturing causal dependencies over statistical correlations. Our work resonates with this principle, i.e., we seek to build an abstract representation of multi-agent interactions that captures the causal relationship between individual agents, as opposed to the spurious correlations in the training data (e.g., agent density vs. interaction intensity).
> * Difference: Unlike previous studies that often focus on identifying causal variables (and structures) in the data generating process, we relax the strict requirement of identifiability and instead place more emphasis on its practical implications, e.g., our proposed sim-to-real causal transfer framework, which seeks to mitigate the absence of real-world causal annotations.
> * Reference: We appreciate the reference that you suggested and enjoyed reading these papers on causal representation learning for domain adaptation. Nevertheless, given that our work is positioned in the multi-agent context with an emphasis on out-of-distribution generalization, we believe that the literature cited in our related work section is more directly pertinent to the main focus of our work.
>
> > Data Sufficiency for Experiments: simulating the trajectories could be helpful in this aspect, such as CausalCity.
>
> * Yes, we fully agree. In fact, this perspective has been integral to the curation of our diagnostic dataset, for which we intervened one agent at a time to obtain counterfactual data pairs, as detailed in Section 3.3.
> * While our work opted for ORCA as a minimalist yet effective simulator of multi-agent interactions, we acknowledged the potential benefits of using more advanced simulators like CausalCity in future research, as discussed in Appendix C.

---

> > ### Public Comment · ~Yuke_Li1 · 2023-11-23
> >
> > I sincerely appreciate your response in the midst of a busy schedule. If possible, I am keen to engage in further discussion on several key points, since they could significantly alleviate concerns about the potential overstatement of "causality" in your work.:
> >
> > - **Assumptions:** While I appreciate your response regarding the assumptions of your paper, I feel there's a need for more clarity. It appears that the paper posits two main assumptions: 1) The interactions among agents exhibit some form of causality; 2) Causal sufficiency is assumed, given the absence of defined confounders. However, it seems that neither of these assumptions has been explicitly justified in the paper.
> >
> > - **Causal Representation Learning:** It's my understanding that causal representation learning is founded on specific hypotheses, like the independent noise condition, and the references offer identifiability results for establishing causal relationships. However, the identifiability concept in causal representation, crucial for theoretical disentanglement, seems to differ from what is proposed in your work. I am uncertain if this aspect has been appropriately adapted in your study. Moreover, references [3,4,5] do not focus on domain adaptation but rather develop a framework for causal representation learning in time-series data.
> >
> > - **Data:** If the objective of this work is to model causality in agent interactions, I would appreciate further clarification on how real-world scenarios like those in the ETH / UCY datasets are utilized in the absence of interventional data.

---

> > > ### Author Response · Authors · 2023-11-23
> > >
> > > Dear Dr. Yuke Li,
> > >
> > > Thank you for your additional comments.
> > >
> > > * Assumptions: we did not assume causal sufficiency. To our knowledge, establishing causal sufficiency *conclusively* in our considered context is highly challenging, if not impossible.
> > > * Data: as answered above, while the interventional data is often absent in the real-world data, it can be gathered from the simulation counterpart -- the core insight behind our proposed sim-to-real causal transfer.
> > > * Reference: our manuscript has cited a rich array of literature in causal learning related to the multi-agent setting, e.g., [A-D], as well as some work, e.g,. [E], related and preceding to your suggested ones [5]. We would gladly discuss the relation between our work and some of your suggested references in our updated manuscript.
> > >
> > > We cannot provide a more detailed response as of now, unfortunately, given that the rebuttal deadline is approaching soon. Nevertheless, we hope our brief response partly addresses your questions, and we welcome any additional feedback you may have.
> > >
> > > [A] Human Trajectory Prediction via Counterfactual Analysis. CVPR, 2021. \
> > > [B] Analyzing Feature Attribution in Trajectory Prediction. ICLR, 2022. \
> > > [C] Causal-based Time Series Domain Generalization for Vehicle Intention Prediction. ICRA, 2022. \
> > > [D] Generative Causal Representation Learning for Out-of-Distribution Motion Forecasting, ICML, 2023. \
> > > [E] AdaRL: What, Where, and How to Adapt in Transfer Reinforcement Learning, ICRL, 2022.

---

### Meta-Review · Area_Chair_1XHa · 2023-12-15

**Metareview:**

The paper proposes to study social trajectory modelling and forecasting under a causal representation lens. Reviewers appreciate the effort, however, they raise significant concerns regarding assumptions and thereafter limitations. For one, the paper reports results on a single dataset only, a remark brought up by reviewer 1Bm7, and as far as I can see, not addressed. While I understand it might be extra effort to extend results to different datasets, I believe we are past the stage that such limited empirical verification is sufficient. Importantly, the reviewers raise significant points on whether considering one pedestrian/agent (independently) at a time is good enough, considering that the causal effects may happen in a group setting instead. The response was that this is easier computationally. However, considering that the work specifically wants to address misaligned previous assumptions, it does not make sense to do so by adding new types of bias. Last, there was concern also regarding whether the causal effects are modelled rigorously and if that should be core to the study, considering also prior works. The rebuttal re-focused on that the paper's contribution is on identifying issues with evaluation, however, this is paradoxical to the remark above regarding considering pedestrians independently. I acknowledge these limitations are hard to address in a short time, thus, I recommend the authors to work further on their manuscript and resubmit.

**Justification For Why Not Higher Score:**

Lack of clarity on the contributions.

**Justification For Why Not Lower Score:**

See above.

---

### Decision · Program_Chairs · 2024-01-16

Reject